# Dendritic Cells: Behind the Scenes of T-Cell Infiltration into the Tumor Microenvironment

**DOI:** 10.3390/cancers13030433

**Published:** 2021-01-23

**Authors:** Valeria Lucarini, Ombretta Melaiu, Patrizia Tempora, Silvia D’Amico, Franco Locatelli, Doriana Fruci

**Affiliations:** 1Department of Paediatric Haematology/Oncology and of Cell and Gene Therapy, Ospedale Pediatrico Bambino Gesù, IRCCS, 00146 Rome, Italy; valeria.lucarini@opbg.net (V.L.); ombretta.melaiu@opbg.net (O.M.); patrizia.tempora@opbg.net (P.T.); silvia.damico@opbg.net (S.D.); franco.locatelli@opbg.net (F.L.); 2Department of Pediatrics, Sapienza University of Rome, 00161 Rome, Italy

**Keywords:** solid tumors, CD8^+^ T-cells, dendritic cells, DC-NK cell axis

## Abstract

**Simple Summary:**

High T-cell infiltration has been associated with improved clinical outcomes in many human solid tumors. However, these cells are not autonomous in their effector function, depending on the interaction with other immune and non-immune cells, as well as soluble factors released into the tumor microenvironment (TME). Identification of the key elements underlying T-cell recruitment within tumors is of fundamental importance to improve the success of immunotherapy strategies. This review summarizes the most recent findings on dendritic cells (DC), a key cellular element that regulates the recruitment of functional tumor-specific CD8^+^ T cells, and current strategies that exploit this innate immune cell to improve the efficacy of therapeutic treatments.

**Abstract:**

Tumor-infiltrating CD8^+^ T cells have been shown to play a crucial role in controlling tumor progression. However, the recruitment and activation of these immune cells at the tumor site are strictly dependent on several factors, including the presence of dendritic cells (DCs), the main orchestrators of the antitumor immune responses. Among the various DC subsets, the role of cDC1s has been demonstrated in several preclinical experimental mouse models. In addition, the high density of tumor-infiltrating cDC1s has been associated with improved survival in many cancer patients. The ability of cDC1s to modulate antitumor activity depends on their interaction with other immune populations, such as NK cells. This evidence has led to the development of new strategies aimed at increasing the abundance and activity of cDC1s in tumors, thus providing attractive new avenues to enhance antitumor immunity for both established and novel anticancer immunotherapies. In this review, we provide an overview of the various subsets of DCs, focusing in particular on the role of cDC1s, their ability to interact with other intratumoral immune cells, and their prognostic significance on solid tumors. Finally, we outline key therapeutic strategies that promote the immunogenic functions of DCs in cancer immunotherapy.

## 1. Introduction

Tumor-infiltrating lymphocytes (TILs) are key elements of the tumor microenvironment (TME), but their presence does not necessarily imply effective induction of anti-tumor immunity. This is because most TILs are not tumor-specific CD8^+^ T cells. Identification of factors that drive recruitment and activation of tumor-specific CD8^+^ T cells is crucial to improve the efficacy of anti-cancer immunity.

Gene expression profiles of metastatic tumors have recently revealed the presence of chemokine transcripts critical for the recruitment of tumor-specific CD8^+^ T cells [1,2,3]. They include chemokines, such as CCL2, CCL3, CCL4, CCL5, CXCL9 and CXCL10, produced by a variety of cell types in the TME, including tumor cells, dendritic cells (DC) and macrophages.

Tumor-specific CD8^+^ T cell activation is a tightly controlled phenomenon that requires initiation signals provided primarily by DCs. The presence of DCs within the tumor is therefore essential to support CD8^+^ T-cell responses against cancer cells. Consistently, in many types of tumors, the presence of activated DCs, as measured by DC gene signatures, is positively correlated with inflammatory status and response to PD-1/PD-L1 pathway inhibition. Thus, understanding how DCs reach tumors and how they are activated is critical for improving immunity against cancer. In this review, we will summarize recent findings on the role of DCs in the recruitment of tumor-specific CD8^+^ T cells and current strategies that exploit these elements to improve the efficacy of cancer immunotherapy.

## 2. DC Cell Subsets in Cancer Immunology

DCs are the most important antigen-presenting cells (APCs), able of initiating and regulating the adaptive immune response [4]. Although their primary function is to capture, process and present antigens to T lymphocytes via major histocompatibility complex class I (MHC-I) and class II (MHC-II) molecules, DCs are able of secreting a wide variety of cytokines and growth factors that are responsible for attracting and interacting with other immune cell populations [5] (Figure 1).

DCs can exist in two different functional states, “mature” and “immature”, which influence the quality of immune responses. Under homeostatic conditions, mature DCs reside in peripheral tissues and acquire the ability to activate naive antigen-specific T cells in secondary lymphoid organs [6,7]. In contrast, immature DCs maintain peripheral tolerance to autoantigens by creating an immunosuppressive environment that inhibits autoreactive T-cell activity and promotes proliferation of regulatory T cells (Tregs) [8,9,10]. Unlike mature DCs, immature DCs express low levels of MHC-I and MHC-II molecules and do not secrete proinflammatory cytokines [10]. Maturation of DCs is triggered by alterations in tissue homeostasis detected by recognition of pathogen-associated molecular patterns (PAMP) or damage-associated molecular patterns (DAMP) [11,12]. Mature DCs migrate to secondary lymphoid organs, where antigen presentation to T cells can occur [13,14]. After maturation, DCs upregulate their antigen presentation machinery and co-stimulatory molecules, such as CD40, CD80 or CD86, becoming potent T-cell activators [15]. Given their importance in mediating T-cell activation, DCs play a key role in the antitumor response [16]. DCs infiltrate most tumors, and the importance of their protective role has emerged over the years. Specifically, DCs engulf, process and present tumor-associated T-cell antigens triggering an antitumor response [17]. Generally, extracellular antigens are internalized by APCs, degraded in the endo/lysosome compartment, and presented to CD4^+^ T cells by MHC-II molecules. In contrast, cytosolic antigens generated during viral infection are processed and presented by MHC-I molecule. Some DC subsets have unique mechanisms for cross-presentation of extracellular antigens by MHC-I molecules to induce potent CD8^+^ T-cell responses [18,19]. This phenomenon is particularly important for tumor immunity. The TME is infiltrated by different DC subsets with various stages of maturation.

DCs are classified into conventional or classical DCs (cDC), plasmacytoid DCs (pDC) and monocyte-derived DCs (moDC). cDCs and pDCs are present and active under steady-state conditions, whereas moDCs arise only during inflammation [20].

### 2.1. cDCs

cDCs are functionally distinct in two subsets: cDC1s able of presenting and cross-presenting both endogenous and exogenous antigens, and cDC2s presenting only exogenous antigens [21]. cDC1s are phenotypically defined by the expression of integrin-αX (CD11c) and MHC-II molecules. Human and murine cDC1s differ in the expression of various markers (Table 1). Murine cDC1s express CD11c, MHC-II, CD103, CD8α, XCR1, CLEC9A and DNGR1 [17] and are developmentally dependent on IRF8, ID2, and BATF3 [22]. Human cDC1s express CD11c, HLA-DR, XCR1, CLEC9A, DNGR1, and CD141. Although cDC1s are the rarest subset of DCs, they play a key role in generating immunity to cancer. Indeed, cDC1s capture apoptotic tumor cells, migrate to draining lymph nodes, and cross-present tumor antigens to CD8^+^ T cells [23,24]. The importance of cross-presentation of cDC1s in cancer immunity has been demonstrated in several mouse models. Mice with impaired cross-presentation, such as mice deficient in DC-specific Sec22b or Wdfy4, are unable to mount antitumor responses and reject tumors [25]. These data clearly document that cross-presentation of cDC1s is crucial for inducing a potent antitumor response. In mouse models, cDC1s are present as lymph node resident (CD8α^+^) and migratory (CD103^+^) populations. The migratory DC population transports antigens from peripheral tissues to the lymph nodes and spleen in a CCR7-dependent manner [24,26]. A substantial fraction of intratumoral CD103^+^ cDC1s does not migrate to the lymph nodes, but still plays a crucial role in cancer immunity. Broz et al. demonstrated that, in mouse models, non-migrating CD103^+^ cDC1s mediate their effects directly in the TME by producing distinct chemokines [27]. The crucial role of CD103^+^ cDC1s was further demonstrated in Batf3^−/−^ mice lacking CD103^+^ cDC1s, which fail to reject tumors and do not respond to immune checkpoint inhibition [28,29]. Given the importance of cDC1s in cancer immunity, several authors have focused their efforts on attempting to recall cDC1s within the tumor site, thereby promoting subsequent CD8^+^ T cell recruitment and efficient control of tumor growth [30,31].

Unlike cDC1s, the role of cDC2s in cancer immunity is less defined. cDC2s are the most important subset of human DCs in blood, lymphoid organs and non-lymphoid tissues, as well as the most efficient APCs for CD4^+^ T-cell activation and expansion [32]. cDC2s also express different markers in mouse and human (Table 1). Murine cDC2s express CD11c, MHC-II, CD11b and CD172a, whereas human cDC2s are characterized by the expression of CD11c, HLA-DR, CD1c, CD1a and CD172a. Compared with cDC1s, cDC2s are superior in inducing CD4^+^ T cell responses through antigen presentation on MHC-II, and in activating Th17 cells, a population with controversial roles in cancer that produces high levels of proinflammatory cytokines [33]. cDC2 are associated with good prognosis in many human cancers. Recently, Binnewies and colleagues demonstrated the crucial role of cDC2s in supporting the antitumor effect of CD4^+^ T cells and response to anti-PD-1 therapy [34]. Indeed, CD4^+^ T-cell responses are significant against tumors. Although CD8^+^ T cells are a very powerful tool in fighting cancer, the help of CD4^+^ T cells is also crucial in establishing efficient and long-lasting cytotoxic CD8^+^ T-cell responses. Moreover, activated CD4^+^ T cells also contribute to antitumor immunity through the production of type II interferon (IFN-γ), which activates Natural Killer (NK) cells and macrophages, inhibits angiogenesis, regulates the generation of tumor stroma, and promotes direct cytolytic effects [35].

### 2.2. pDCs

pDCs are characterized by different markers in mice and human (Table 1) [36]. This DC subset is primarily found in lymphoid organs and can migrate through the bloodstream into lymph nodes. pDCs act like APCs, but less efficiently than cDC1s and cDC2s [37]. Although pDCs are a subset of DCs studied in relation to viral infections and autoimmune diseases, they also play an important role in cancer, although this is still controversial. pDCs are potent producers of type I interferon (IFN-α/β) which inhibits tumor cell proliferation, angiogenesis and metastasis [38]. Moreover, in mouse models of breast cancer, pDCs have direct cytotoxic activity mediated TRAIL and Granzyme B expression, both in vitro and in vivo [39,40]. On the other hand, pDCs are known to induce immune tolerance and promote tumor growth. In human melanoma, ovarian and breast cancers, pDCs contribute to the generation of an immunosuppressive TME that supports tumor progression by reducing the cytotoxic activity of CD8^+^ T cells and increasing the recruitment of Foxp3^+^ Tregs that produce IL-10 and TGF-β. Therefore, the presence of pDCs in the TME of these tumors is associated with a poor prognosis [41,42]. In addition, other studies have reported the induction of angiogenesis by several cytokines produced by pDCs, such as TNFα, IL-8 and IL-1α [43,44].

### 2.3. MoDCs

MoDCs are another subset of DCs generated by monocytes in an inflammatory environment. These cells are characterized by the expression of cDC2-like markers. Murine moDCs express CD11c, MHC-II, CD11b, Ly6C, CD14, CD64, CD206, CD209 and CCR2, while human moDC additionally express CD1c and CD1a (Table 1).

The heterogeneity of moDCs may explain the contradictory roles observed in many types of cancers. Notably, moDCs are primarily generated in response to inflammation and promote CD4^+^ T-cell differentiation towards a Th1, Th2 or Th17 cell phenotype [45]. They are also effective in cross-presenting tumor antigens towards CD8^+^ T cells and inducing their infiltration into the TME of different mouse tumor models [46]. Conversely, they may exhibit an immunosuppressive phenotype that expresses high levels of iNOS, TNFα, IL-6, IL-10, and has the ability to hinder T-cell proliferation [47].

## 3. Chemokine Networks and DCs in the TME

Chemokines are key regulators of immune cell trafficking in the TME. Many chemokines drive DC migration to primary and secondary lymphoid organs and to peripheral tissues. Circulating cDC1s are recruited to the TME in response to CCL5 and XCL1 produced by different immune cell populations, such as CD8^+^ T cells, NK cells and innate lymphoid cells [48]. However, while CCL5 can also recruit tumor-promoting cells, such as macrophages and Tregs [49], XCL1 is a specific chemoattractant for cDC1s [50]. Expression of XCR1 is critical for cDC1 functions and for promoting DC migration in response to XCL1 ligand. Indeed, the XCR1/XCL1 axis is essential for the development of efficient cytotoxic CD8^+^ T cells [51]. FLT3L is an important factor produced by intratumoral NK cells that support the viability and functions of cDC1s within the TME, promoting their local differentiation from precursor cells [52]. CCL3 and CCL20 drive the migration of mature and immature DCs to the tumor site [53,54]. Melanoma cells specifically produce CCL20, which recruit circulating immature DCs through the CCR6 receptor [55].

In addition, cDC1s act by promoting immunity to cancer through the production of CXCL9 and CXCL10, two important chemokines responsible for the recruitment of effective T cells to the tumor site [56]. These chemokines are also crucial in positioning CD8^+^ memory T cells in cDC1-rich areas to induce local T-cell restimulation [57,58]. The importance of these chemokines has been demonstrated in mouse models in which the absence of CXCL9 and CXCL10 prevents CD8^+^ T-cell recall within the tumor [56,59]. De Mingo Pulido and colleagues demonstrated that the TIM3 receptor, when expressed by cDCs infiltrating breast cancer, inhibits CD8^+^ T-cell recruitment through downregulation of CXCL9 expression [29]. cDC1 and cDC2 also locally induce IL-12 production and subsequently CD8^+^ T cell cytotoxicity and IFN-α/β production [60]. cDC2s produce a wide variety of cytokines important for CD4^+^ T-cell activation and Th1 and Th2 responses, such as IL-1β, IL-6, IL-12 and IL-23 [45].

In addition to cDCs, pDCs are also important in the TME. Their recruitment is driven by the CXCL12/CXCR4 axis where CXCL12 may act as a survival factor for tumor-infiltrating pDCs [61,62].

TME may also affect the function and stimulation of DCs. DCs may promote tumor growth and progression by enhancing immune tolerance. Within TME, several soluble factors may upregulate transcriptional and metabolic pathways for the generation of the tolerogenic DC phenotype, such as prostaglandin E2 (PGE2), TGF-β, VEGF, IL-10, IL-6, and colony stimulating factor-1 (CSF-1). PGE2, produced by tumor cells, can inhibit IL-12 production by cDC1, downregulate the expression of co-stimulatory molecules, and prevent the induction of antitumor responses [63]. TGF-β and VEGF, produced by the tumor, inhibit DC functions, including differentiation from precursors, activation and recruitment to the tumor site [64,65]. TGF-β not only inhibits DCs, but also regulates the activities of several immune cells, including T cells, macrophages and B cells, and may support Treg cell development [65].

Crosstalk between tumor-associated macrophages and DCs is crucial in the TME. Indeed, in mouse mammary tumors, IL-10 production by macrophages can suppress IL-12 expression by CD103^+^ cDC1s resulting in activation of tumor-specific CD8^+^ T cells [66]. In addition, DCs can produce inhibitor factors. Tumor-derived TLR2 ligands have been shown to be critical for the generation of immunosuppressive IL-6- and IL-10-producing DCs [67]. Moreover, Spranger and colleagues demonstrated that Wnt/β-catenin activation in tumor cells is variously involved in the suppression of DC function by paracrine IL-10 production and CCL4 downregulation [56].

Within the TME, DCs are the main producers of CCL22, a chemokine that regulates Treg cell migration [68,69]. Treg-DC interaction at the tumor site is critical for the local suppressive function of Tregs [69]. The production of indoleamine 2,3 dioxygenase (IDO) is upregulated in tumor-associated DCs, mainly in pDCs, and is responsible for promoting Treg cell differentiation [70].

## 4. DC-NK Cell Axis in Anti-Cancer Immunity

A recent advance in the field of antitumor immunity concerns the reciprocal interplay between DCs and NK cells [71] (Figure 2). Indeed, in addition to direct cytotoxic activity, NK cells are able to modulate antitumor immune responses through interactions with DCs. The first evidence of a bidirectional cross talk between these two innate immune cells comes from the finding that contact between DCs and resting NK cells induces an increase in IFN-γ production and cytolytic activity of NK cells [72]. Accordingly, the antitumor effects triggered by NK cells were significantly reduced in DC-depleted mice [72]. Several mechanisms have been proposed to demonstrate the ability of DCs to affect NK cell function. First, the formation of stimulatory synapses between DCs and NK cells promotes IL-12 release from DCs, which in turn activates IFN-γ secretion from NK cells [73]. Second, bone marrow-derived DCs from wild-type mice are able to activate NK cells to produce IFN-γ in an IL-12-dependent manner. This was not observed in Batf3-deficient mice lacking CD103^+^ DCs, which develop more spontaneous metastases and survive less than wild-type control mice [74]. Third, therapeutic injection of recombinant mouse IL-12 reduced metastases in both wild-type and Batf3-deficient mice, highlighting the role of IL-12 produced by CD103^+^ DCs in controlling NK cell-mediated tumor metastasis [74]. A key role of CX3CL1, a chemokine expressed by mature DCs, has also been demonstrated in the activation of resting NK cells [75]. Another factor that has been shown to be crucial in anti-tumor immunity is IL-15, a cytokine produced by DCs that can promote antitumor activity of NK cells toward both NK-sensitive and NK-resistant targets [76].

Further evidence has shown that interaction of NK cells with immature autologous DCs leads to their mutual activation. Specifically, fresh NK cells cultured with immature DCs in the presence of a specific stimulus strongly enhanced DC maturation and IL-12 production, thereby increasing their ability to stimulate naive CD4^+^ T cells [77]. Another mechanism involves the ability of DCs to activate NK cells by IL-18. Once activated, NK cells release HMGB1, which in turn, promotes inflammation by inducing maturation of DCs [78].

The ability of NK cells to enhance the activity of DCs has recently been investigated proving to be a key element in the stimulation of antitumor T-cell responses. First, several studies have reported that activated NK cells kill autologous immature DCs, both in vitro and in vivo, in favor of fully activated DCs [79,80]. This evidence led to the proposal that NK cell-mediated DC killing might be crucial for promoting the development of a more immunogenic subset of DCs, which can promote the expansion of tumor-specific CD8^+^ T cells. Interestingly, NK cells have been shown to be necessary for the accumulation of cDC1s in mouse tumors, as they produce CCL5 and XCL1, two chemo-attractants of cDC1s that express CCR1, CCR5 and XCR1 receptors [48]. Furthermore, NK cell-derived chemokines regulate the distribution of cDC1s within tumor tissues, allowing them to localize in close proximity to NK cells with which they often form multicellular clusters. It has been hypothesized that this cross-interaction could be further supported by chemokines secreted by cDC1s such as CXCL9 and CXCL10, which can attract NK cells to TME via CXCR3 [48,56,81].

In addition to chemokines, the abundance of DCs within tumors is also regulated by the local availability of DC growth factors, such as FLT3L. Recently, NK cells have been shown to be the major source of FLT3L. Genetic and cellular ablation of NK cells has provided evidence that these cells are essential in regulating DC accumulation within the tumor through the production of FLT3L. A positive correlation was found between the expression of the gene encoding FLT3L and the abundance of DC and NK cells [52]. The existence of the DC-NK cell axis is further corroborated by the presence of stable conjugates between these cells within the TME [52]. Consistent with these findings, treatment of mice with FLT3L resulted in the expansion and accumulation of activated CD103^+^ DC progenitors in melanoma lesions [26]. Similar results were shown in mouse models of breast cancer [30] and pancreatic ductal adenocarcinoma, where treatment with FLT3L plus CD40 agonist elicited integrated antitumor responses that led to a marked increase in tumor-infiltrating NK and NKT cells [31]. To date, evidence for the existence of the DC-NK cell axis comes mainly from studies in mouse tumor models. Since human cDC1s have similar characteristics to mouse cDC1s, both genetically and functionally, it is likely that the same crosstalk may also occur in humans [82]. There is evidence in human tumors to support this hypothesis. A positive association between the transcription levels of CCL5, XCL1, and its paralog XCL2, has been detected in human skin cutaneous melanoma, invasive breast carcinoma, head and neck squamous cell carcinoma and lung adenocarcinoma [48]. Data are consistent with the hypothesis that these chemokines may be produced by intratumoral NK cells, as supported by the strong correlation between their transcripts and NK cell gene signatures [48]. An independent study has shown that the lack of cDC1s correlates with that of intratumoral NK cells in lung adenocarcinoma [83], supporting the hypothesis that NK cells contribute to the accumulation of cDC1s within human TME. Consistent with the report by Bottcher and colleagues [48], Krummel’s group found a significant correlation between FLT3L expression and DC levels as estimated with a specific DC gene signature [27]. The authors also demonstrated that DC abundance correlated with intratumoral NK cell levels, and that both cell types may predict response to anti-PD-1 immunotherapy in melanoma patients. These data indicate that, in humans, intratumoral BDCA3^+^ DC levels are controlled by activated NK cells through FLT3L production [52]. The importance of the DC-NK cell axis highlighted in adult cancers paves the way for new therapies. Interestingly, recent advances in neuroblastoma immune profiling have highlighted the critical role of both DCs and NK cells in establishing the immune-inflamed phenotype [84]. Indeed, it has been shown that neuroblastoma specimens highly infiltrated by T cells are also enriched with intratumoral DCs and NK cells. Melaiu and colleagues, defined two gene signatures related to DCs and NK cells, which are strongly correlated with PD-1 and PD-L1 expression. Interestingly, the identified DC gene signature includes transcripts typically expressed by NK cells, and conversely, the NK gene signature includes the BTLA transcript that is typically expressed by DCs, thus confirming the existence of a DC-NK cell axis also in neuroblastoma. This hypothesis was further strengthened by multiplexed immunofluorescence imaging which clearly shows the interaction between DCs and NK cells, both with each other and with CD8^+^ T cells, within the tumor nests of low-risk neuroblastoma patients [85]. Moreover, the FLT3L and CCL5 genes both strongly correlated with DC, NK cell and T cell abundance, thus further validating the existence of a DC-NK cell axis that promotes CD8^+^ T cell antitumor immunity in this pediatric malignancy. In line with these findings, Belounis and colleagues recently demonstrated that stimulation with NK cells by Toll-like receptor (TLR)-activated pDCs enhances the efficacy of dinutuximab-based immunotherapy by increasing the treatment ability to mediate autologous killing of patient-derived neuroblastoma cells [86]. All these data may be of great help in selecting patients with the greatest chance of benefiting from currently available immunotherapies, which have been extensively reviewed elsewhere [87], as well as in improving efficiency in the treatment of childhood malignancies.

## 5. Prognostic Value of cDC1s in Solid Tumors

High density of tumor-infiltrating cDC1s has been associated with better prognosis in many cancers. Broz and colleagues were among the first to demonstrate the importance of CD103^+^ cDC1 in stimulating tumor-specific CD8^+^ T cell responses within the TME [27]. The authors identified a DC gene signature based on the ratio of CD103^+^/CD103^−^-associated genes that provides a strong pro-immune survival signal in breast cancer, head-neck squamous cell carcinoma and lung adenocarcinoma, suggesting that CD103^+^ cDC1s are critical for robust tumor control in both mice and humans [27]. Querying the TCGA dataset, Bottcher and colleagues found a strong association between high expression of the DC gene signature and improved clinical outcome in patients with skin cutaneous melanoma, invasive breast carcinoma, head and neck squamous cell carcinoma and lung adenocarcinoma [48]. Two other studies have demonstrated the prognostic value of cDC1s in breast cancer. Michea and colleagues used RNA-based next-generation sequencing to systematically analyze the transcripts of all DC subsets and showed a significant association between cDC1s and prolonged survival of patients with luminal breast cancer [88]. Hubert and colleagues demonstrated that human cDC1s play an important role in the antitumor immune response through their ability to produce type III interferon (IFN-λ), which is crucial for promoting a Th1 environment through increased production of IL-12p70, IFN-γ and cytotoxic lymphocyte-recruiting chemokines [89]. Both IFN-λ1 and its receptor have been associated with favorable patient clinical outcome. The authors also demonstrated that TLR3 engagement is able to induce IFN-λ production from tumor-associated cDC1s, thus proposing TLR3 activation as a potential therapeutic strategy. Interestingly, the authors also demonstrated a strong positive prognostic value of cDC1s for patients with colorectal adenocarcinoma, head and neck squamous cell carcinoma, liver hepatocellular carcinoma, kidney renal papillary cell carcinoma, lung adenocarcinoma, skin cutaneous metastatic melanoma and thyroid cancer [89]. The positive prognostic value of cDC1 infiltration in melanoma has been previously demonstrated [90]. Ladányi and colleagues evaluated the maturation status of DCs and showed that a high peritumoral density of mature DC-LAMP^+^ DCs was associated with increased infiltration of activated T lymphocytes and improved survival of melanoma patients [90]. Roberts and colleagues demonstrated that CD103^+^ DCs traffic tumor antigens to lymph nodes in a CCR7-dependent manner and that this trafficking is critical for effective antitumor CD8^+^ T cell priming [24]. The authors also found that intratumoral expression of CCR7 strongly predicts T cell infiltration and overall survival in patients with metastatic melanoma [24]. Barry and colleagues demonstrated that high levels of DCs, in combination with NK cells, lead to both improved response to anti-PD1 immunotherapy and increased survival in melanoma patients [52]. Similar results were obtained in ovarian cancer, where the high density of tumor-infiltrating LAMP^+^ DCs was strongly associated with an immune context characterized by Th1 polarization and cytotoxic activity, as well as with a more favorable overall survival of patients with high-grade serous ovarian carcinoma [91]. In lung cancer, mature DCs were found localized mainly within tertiary lymphoid structures (TLS), known to exacerbate a local immune response. The high density of TLS-associated DCs correlates with long-term survival and high expression of genes related to T-cell activation, Th1 phenotype, and cytotoxic activity [92]. In human neuroblastoma, DCs were sparsely distributed or localized within the TLS, and significantly correlated with the abundance of tumor-infiltrating T cells and NK cells, both at the transcriptional and protein levels, and associated with favorable prognosis [85]. Interestingly, high expression of DC and NK cell-gene signatures were predictors of good prognosis not only in patients with neuroblastoma, but also in patients with colorectal cancer, skin cutaneous melanoma, head and neck squamous cell carcinoma and breast cancer [85], thus representing a robust prognostic tool to be added to those currently used, in addition to the T-cell infiltration score [93,94].

## 6. Clinical Trials Exploiting the Efficacy of Agents and Therapies That Promote the Immunogenic Functions of DCs in Cancer Immunotherapy

Given the importance of cDC1s in cancer immunotherapy, the use of strategies targeting the recruitment, expansion and activation of this subset of DCs in the TME, may contribute to increase antitumor immunity and the success of cancer immunotherapy. Many clinical trials of DC-based anticancer immunotherapy have shown encouraging results, particularly when combined with other therapies aimed at inducing the full maturation and activation of DCs necessary to stimulate tumor-specific CD8^+^ T cell responses. Of great interest are factors that promote the immunogenic functions of DCs, such as ligands for Toll-like receptors (TLRs), the CD40 receptor, as well as cytokines essential for the development and mobilization of DCs in lymphoid organs, peripheral blood and bone marrow [95,96,97]. In addition, many therapies currently in clinical use have been found to induce DC activation and maturation by promoting and enhancing CD8-dependent immune responses in poorly immunogenic tumors. These include chemotherapy, radiation therapy, irreversible electroporation and cryoablation. Finally, particular interesting are the recent evidences on the unique role of DCs in the PD-L1-PD1 regulatory axis and in the regulation of antitumor immunity.

### 6.1. Agents Promoting the Immunogenic Functions of DCs

TLR signaling causes DCs to mature, resulting in expression of MHC class I and II molecules and secretion of pro-inflammatory cytokines. The use of TLR ligands has been correlated with maintaining DCs in an active state to induce immune responses. Human CD141^+^ DCs express high levels of TLR3, and treatment with TLR3 agonists, such as polyinosinic-polycytidylic acid (Poly I:C) and its derivates Poly-ICLC (Hiltonol) and poly-IC12U (Ampligen), have been shown to efficiently activate DCs by inducing cross-presentation, pro-inflammatory cytokines, Th1 cell immunity, NK cells and cytotoxic CD8^+^ T cell responses [26,98,99]. Intratumoral administration of Hiltonol and Ampligen, in combination with other therapies, has been approved for the treatment of melanoma, glioma, head and neck squamous cell carcinoma, breast and ovarian cancers in eight ongoing Phase I/II clinical trials (Table 2). Several clinical trials (Phase I/II) with TLR7/8 agonists, such as Imiquimod, in combination with other therapies are ongoing in different cancer types (Table 2). Local treatment with TLR7/8 agonists activates all DC subsets and induces NF-kB, pro-inflammatory cytokines and costimulatory receptors [97]. Several TLR9 ligands able of activating DCs in vivo are also being investigated in combination with other therapies, including immune checkpoint inhibitors (Table 2) [95,100].

The CD40 receptor is a promising target. Increased binding between CD40 and its ligand CD40L, expressed on CD4^+^ T cells, results in increased co-stimulatory molecules and cytokine production, and subsequent activation of CD8^+^ T cells. The interaction between CD40 and CD40L may also increase cross-presentation by DCs. The use of a recombinant receptor containing the cytoplasmic domain of CD40 is a novel method to amplify DC activation. This domain is combined with ligand-binding domains and a membrane targeting sequence. Thus, activation of DCs with this recombinant receptor promotes the induction of CD8^+^ T cells that could cause tumor cell eradication.

FLT3L binds DCs by inducing their proliferation, differentiation, development and mobilization. Administration of FLT3L in combination with radiotherapy to promote immunogenic tumor cell death and maturation of DCs, and dual TLR3/CD40 stimulation to activate antigen-loaded cDC1 for priming and expansion of tumor-specific CD8^+^ T cells, has been shown to enhance tumor immunity in several mouse models [30]. These encouraging results have led the Food and Drug Administration (FDA) to approve the use of human recombinant FLT3L (CDX-301) in the clinic for multiple tumors. A Phase I study initially demonstrated that CDX-301 had an acceptable safety profile and could mobilize DCs in healthy volunteers [101]. An ongoing Phase II study is planned to evaluate the combination of CDX-301 with stereotactic body radiation therapy directed at single tumor lesions in patients with advanced non-small cell lung cancer (NCT02839265). An in-situ vaccination strategy combining FLT3L, local radiotherapy, and a TLR3 agonist (poly-ICLC), has been shown to be feasible, safe and able to induce recruitment of antigen-loaded and intratumoral cross-presenting DCs, resulting in the generation of a systemic tumor-specific CD8^+^ T cell response and regression of both primary and untreated distant tumors in patients with non-Hodgkin’s lymphoma (NCT01976585) [102]. This clinical trial was followed by others currently on going in patients with breast cancer and head and neck squamous cell carcinoma (NCT03789097), as well as with non-small cell lung cancer and lung cancer (NCT04491084) (Table 2). The addition of anti-CD40 antibody to FLT3L, radiotherapy and Poly-ICLC has been shown to induce regression of poorly T cell-infiltrated tumors refractory to PD-1/PD-L1 therapy in four syngeneic mouse models: colon adenocarcinoma, melanoma and two triple-negative mammary cancers of melanoma [30]. Interestingly, this treatment increases the infiltration of CD8^+^ T cells that mediate the regression not only of primary, but also of untreated distant tumors poorly infiltrated by T cells [30]. These very encouraging results were followed by a Phase I clinical trial initiated on December 1th, 2020 to evaluate the safety of in situ immunomodulation with recombinant CDX-301, radiotherapy, anti-CD40 mAb and Poly-ICLC in patients with unresectable and metastatic breast cancer (NCT04616248).

### 6.2. DCs and Chemotherapy 

Many chemotherapeutic agents are known to promote antitumor activity by triggering immunogenic cell death (ICD) of tumor cells [103,104]. ICD begins with the induction of cellular stress and culminates in cell death following exposure to and release of several DAMPs. ICD-associated DAMPs include surface-exposed calreticulin as well as secreted ATP, annexin-1, type I interferon, and high-mobility group box 1 (HMGB1). Binding of DAMPs to specific DC-expressed receptors initiates a cascade of events that ends with activation of innate and adaptive immune responses. Clinical and preclinical evidence indicates that DAMPs may have prognostic and predictive values for cancer patients. DAMPs are capable of eliciting anticancer immune responses to enhance the therapeutic effects of conventional chemotherapy and radiation therapy. However, only few ICD-inducing agents have been successfully used in the clinic as therapy [105]. Clinical and preclinical studies indicate that these agents may be particularly relevant to the activation of antitumor immune responses triggered by ICI or other forms of immunotherapy in the context of combinatorial treatment regimens. In this regard, a number of FDA-approved ICD inducers are currently being studied in the oncology setting, especially in combination with immunotherapeutic agents [106].

### 6.3. DCs and Radiation Therapy

Radiation therapy causes cell death in highly replicating cancer cells by creating breaks in the DNA double-strand. Several evidences indicate that radiation therapy has an abscopal effect causing cancer regression in non-irradiated metastatic sites [107]. Like chemotherapy, radiation therapy triggers ICD by causing the release of DAMPs and tumor-associated antigens and thereby priming tumor-specific immunity [106]. Cytosolic DNA released from tumor cells after radiation therapy acts as a DAMP by inducing the production of type I interferon from DCs via cGAS/STING. This pathway is activated on double-stranded DNA (dsDNA) that binds to cGAS (cyclic guanosine monophosphate-adenosine monophosphate (cGAMP) synthase, cGAS) [108,109]. cGAMP functions as a second messenger and binds STING to promote TANK-binding kinase (TAK1)-dependent signal transduction cascade that initiates IRF3- and NF-kB-dependent transcription and culminates in secretion of cytokines that recall cDC1s, including type I IFN, IL-6, TNF and CCL5 [110,111]. Given the importance of type I IFN in early anticancer immune responses, several STING agonists have been tested in clinical trials with the rationale of activating STING in tumor cells or tumor-infiltrating immune cells, including DCs, to achieve immunostimulatory effects alone or in combination with a number of established chemotherapeutic and immunotherapeutic regimens that directly activate STING [112].

### 6.4. DCs and Irreversible Electroporation 

Irreversible electroporation (IRE) is a new ablative technology that uses high-voltage electrical pulses to induce ICD through permanent membrane lysis or loss of homeostasis. The use of IRE for tumor ablation was recently introduced by Jiang and colleagues [113]. This process preserves adjacent structures by contributing to DC activation and maturation as well as immune cell infiltration. An increase in T-cell levels after IRE has been reported in several mouse models [114,115]. Recently, the combination of IRE and anti-PD1 treatment has been shown to promote selective tumor infiltration by CD8^+^ T cells, and to significantly suppress tumor growth and prolong survival of immunocompetent mice bearing pancreatic cancer and melanoma [116].

### 6.5. DCs and Cryoablation

Cryoablation induces tumor cell death by necrosis and osmosis. In the process of necrosis, the intracellular contents of damaged tumor cells are preserved, while DNA, RNA and heat shock protein are released. These agents can induce danger signals, which are able to mature DCs to fully activate T cells, which can lead to a specific immune response [117]. In contrast, cells in the outer margin of cryoablated tissue die by apoptosis, while DNA, RNA and HSPs are preserved. DCs without a danger signal remain immature and thus unable to activate T cells. Therefore, cryoablation can induce both an immunostimulatory and immunosuppressive response. Cryoablation also induces the release of immunosuppressive and immunostimulatory cytokines. In liver tumors, cryoablation in more than 20% of the tissue causes a systemic inflammatory response due to the release of IL-6, IL-10 and TNFα [118]. Two clinical trials reported improved overall survival by combining cryoablation with infusion of allogenic NK cells and DC-activated cytokine-induced killer cells (DC-CIK) in non-small cell lung cancer, respectively [119,120]. The preclinical study by den Brok showed that cryoablation leads to maturation of DCs, which resulted in a tumor-specific immune response that protected half of the mice from a new infusion of tumor cells. This antitumor effect was further enhanced (up to 80%) when combined with administration of the CTLA-4 antibody or Treg-cell depletion [121]. Several immunotherapies can be used to enhance the immunogenic effect of cryoablation as recently reviewed [117,122,123].

### 6.6. PD-L1 on DCs and Immune Checkpoint Blockade Therapies

Immune checkpoint blockade (ICB) therapies have shown clinical promise in a variety of human cancers, being able to restore the anti-tumor immunity response. The significant contribution of the cDC1 subset in determining response to ICB therapy was recently reviewed by Wculek and colleague [16]. Noteworthy is the newly discovered role of DCs in the PD-L1/PD-1 axis and in the regulation of anti-cancer immune responses.

Assays that measure PD-L1 expression on tumor cells has been approved by the FDA as a useful biomarker to determine whether a patient may benefit from checkpoint blockade therapy. However, the fact that nearly half of patients with PD-L1-positive tumors do not respond, whereas some patients with PD-L1 negative tumors may still respond to PD-L1 blockade, suggests the existence of more complex mechanisms than originally presumed. Recent observations have highlighted the functional importance of PD-L1- expressing immune cells, particularly DCs. Peng and colleagues [124], found that DCs upregulate PD-L1 upon antigen uptake, following the production of type II interferon by CD8^+^ T cells. Expression of PD-L1 on DCs correlate with good prognosis and CD8^+^ T cell infiltration in colon cancer [125], and it is essential to protect them from killing by cytotoxic T cells [124], thereby dampening antitumor immune responses. Indeed, the therapeutic effects of PD-L1 blockade disappeared completely in cDC1-deficient mice, even in the presence of other cells with high levels of PD-L1 [124]. Similarly, Oh and colleagues demonstrate the importance of PD-L1 expressed by DCs for antitumor immunity, compared with that expressed by other immune cells of the myeloid lineage [126]. Notably, deletion of PD-L1 in DCs, but not in macrophages, significantly reduced tumor growth and led to enhanced antitumor CD8^+^ T-cell responses [126]. Several authors [124,127,128,129,130] have shown that PD-L1 expressed on DCs is also able to bind in cis CD80, a key costimulatory molecule expressed by DCs [127]. Mayoux et al., demonstrated that the PD-L1/CD80 cis interaction traps PD-L1 preventing its binding to PD-1 and the subsequent inhibition of T-cell function [131]. The authors showed that blocking PD-L1 on DCs relieves CD80 sequestration in cis by PD-L1, which allows the CD80/CD28 interaction to stimulate allogeneic T cell proliferation and enhance T cell priming. The contribution of tumor-associated DCs to anti-PD-L1 therapies was further supported by the fact that patients with renal cell carcinoma or lung cancer treated with PD-L1 blockade (atezolizumab) experienced clinical benefit in the presence of high level of the DC gene signature, thus highlighting the importance of detecting the amount of DCs to support decision-making on treatment option [131]. These recent findings suggest that ICB therapy is effective not only by directly activating T cells, but also by triggering a complex network, in which DCs play a pivotal role at the interface between innate and adaptive antitumor responses [132], and that blocking the PD-1 pathway early with immune checkpoint inhibitors, at the time of priming and expansion of memory T cells, could be critical for enhancing antitumor immunity.

## 7. Ex Vivo Manipulation of cDC1s in Cancer Immunotherapy

In addition to in vivo modulation of cDC1s, ex vivo manipulation is also yielding encouraging results for clinical purposes. Due to the scarcity of cDC1s in peripheral blood, representing only 0.03% of human PBMCs, great efforts have been made to optimize novel protocols for their ex vivo differentiation. Poulin and colleagues identified a method to obtain functional CD141^+^ DCs from in vitro expanded cord blood CD34^+^ precursors by culturing them in medium supplemented with SCF, GM-CSF, IL4 and FLT3L [133,134,135,136]. These cells are able to internalize material from dead cells and cross-presenting processed antigens to CD8^+^ T cells [133]. Other authors obtain a large number of functional cDC1s from cord blood or GM-CSF-mobilized blood CD34^+^ cells by inhibiting the aryl hydrocarbon receptor with its antagonist StemRegenin 1 (SR1) [137]. Other methods consist in obtaining a population of functional CD141^+^ DCs from moDCs exposed to mycolic acid and/or lipoarabinomannan or induced pluripotent stem cells (iPSCs) [138,139].

Strategies currently evaluated in clinical trials also include the increasing XCL1 expression within the tumor to recruit CD141^+^ XCR1^+^ DCs. The safety and immunological effect of genetically modified neuroblastoma cells to express XCL1 and IL-2 were evaluated in two Phase I clinical trials (NCT00062855 and NCT01713439). The results indicated an improved immune response at the tumor site leading to more effective tumor cell killing in relapsed/refractory neuroblastoma [140,141]. The promising results allowed the initiation of another Phase I/II clinical trial (NCT00703222), which is currently ongoing, to evaluate the safety and immunological and clinical monitoring of the efficacy of combined neuroblastoma cell administration. Another ongoing Phase I/II clinical trial (NCT01192555) focuses on the administration of a biologic vaccine using the same neuroblastoma cell line that produces XCL1 and IL-2 in combination with cyclophosphamide, with the aim of further preventing incomplete elimination and recurrence of high-risk neuroblastoma.

## 8. Conclusions

T-cell infiltration in the TME is a critical determinant of response to immunotherapy [142]. The ability of cDC1s to remodel the type, density and repertoire of intratumoral T lymphocytes by converting poorly T cell-infiltrated tumors to T cell-inflamed, has encouraged investigations to identify rationally combined treatment regimens to induce and activate these cells in situ in patients with poorly T cell-infiltrated tumors refractory to anti PD-1/PD-L1 therapy. Recent evidence also showed that cDC1s engaged other immune cell types, including NK cells, establishing intricate immune cross-talks within the TME. The DC-NK axis involvement in mediating the recruitment of tumor-specific CD8^+^ T cells within the tumor site has been shown to be of great importance in promoting improved survival of both adult and pediatric patients. These results pave the way for the exploitation of new immunotherapeutic strategies that, in addition to enhancing the activity of T cells, look behind the scenes, orchestrating the action of all those cellular actors that, together with DCs, are crucial to fight cancer progression.

## Figures and Tables

**Figure 1 cancers-13-00433-f001:**
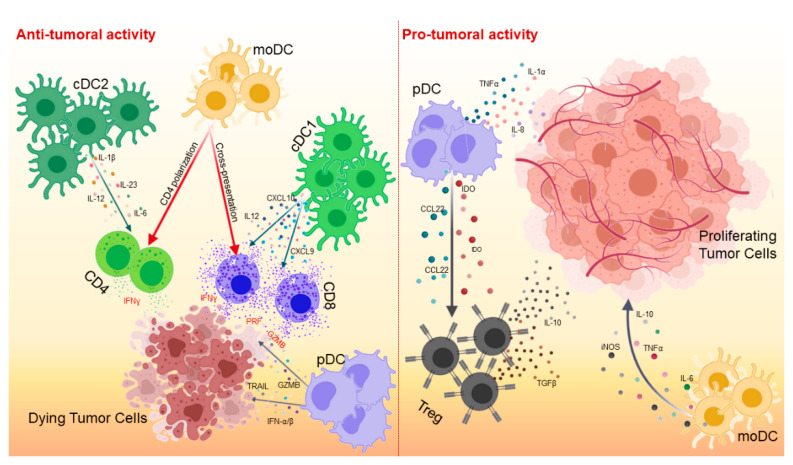
Activity of dendritic cell (DC) subsets in cancer immunity. In the TME, DCs can have both an anti-tumor and pro-tumor effect. Anti-tumor activity is mainly driven by cDC1s and cDC2s (left panel). cDC1s induce recruitment and activation of CD8^+^ T lymphocytes in the TME through cytokine production and cross-presentation of tumor antigens, respectively. cDC2s are the major activators of CD4^+^ T cells. moDCs act mainly by stimulating cDC1s and cDC2s, whereas pDCs kill tumor cells through the expression of IFN-α/β, TRAIL and Granzyme B (GZMB). moDCs and pDCs may also have a pro-tumor action by creating an immunosuppressive environment and promoting tumor growth (right panel). moDCs produce molecules with immunosuppressive function such as iNOS, TNFα, IL-6 and IL-10. pDCs secrete chemokines able to recruit Tregs into the TME (CCL22 and IDO) as well as pro-angiogenic cytokines (TNFα, IL-8, and IL-1α).

**Figure 2 cancers-13-00433-f002:**
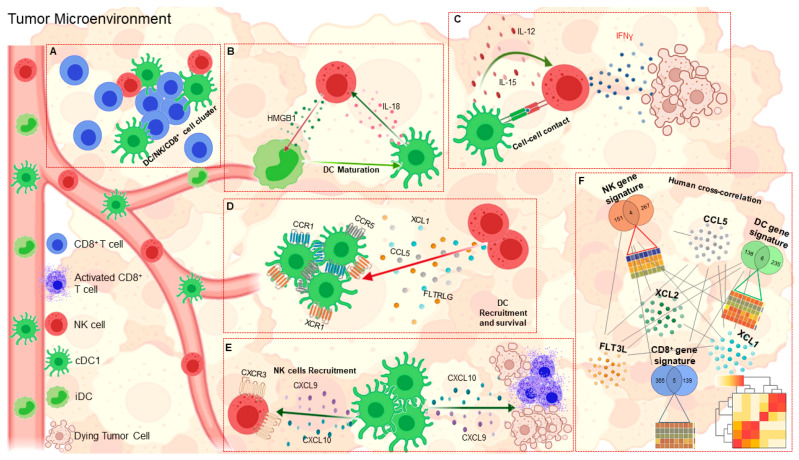
DCs-NK cells cross-talk in cancer immunity. (**A**), DCs, NK cells and CD8^+^ T cells interact with each other in the TME. (**B**), Loop of maturation of DCs by HMGB1 produced by NK cells and activation of NK cells by IL-18 released by DCs. (**C**), Ability of cDCs to increase IFN-γ production by NK cells and amplify their anti-tumor activity either by a cell-cell contact mechanism or by release of the cytokines IL-12 and IL-15. (**D**), Recruitment and survival of cDCs are dependent on the chemokines CCL5, XCL1/2 and FLT3L produced by NK cells. (**E**), CXCL9 and CXCL10 chemokines released by DCs are crucial for both recruitment and activation of NK cells and CD8^+^ T cells in the TME. (**F**), Significant correlations between the genetic signatures of cDC1s, NK cells and CD8^+^ T cells and the key molecules involved in the interaction between cDC1s and NK cells are essential for stratifying low- and high-risk cancer patients.

**Table 1 cancers-13-00433-t001:** Mouse and human DC markers.

DC Subset	Mouse Surface Markers	Human Surface Markers
cDC1	CD11c^+^	CD11c ^low^
	MHC-II^+^	HLA-DR^+^
	CD103^+^	CD141^+^
	XCR1^+^	XCR1^+^
	CLEC9A^+^	CLEC9A^+^
	DEC205^+^	DEC205^+^
	CD8α^+^	
cDC2	CD11c^+^	CD11c^+^
	MHC-II^+^	HLA-DR^+^
	CD172a^+^	CD172a^+^
	CD11b^+^	CD1a^+^
		CD1c^+^
pDC	CD11c ^low^	CD11c^−^
	MHC-II ^low^	HLA-DR ^low^
	CXCR3^+^	CXCR3^+^
	CD317^+^	CD123^+^
	SIGLEC-H^+^	CD303^+^
	B220^+^	CD304^+^
moDC	CD11c^+^	CD11c^+^
	MHC-II^+^	HLA-DR^+^
	CD11b^+^	CD11b^+^
	CD14^+^	CD14^+^
	CD64^+^	CD64^+^
	CD206^+^	CD206^+^
	CD209^+^	CD209^+^
	CCR2^+^	CCR2^+^
	Ly6C^+^	CD1a^+^
		CD1c^+^

**Table 2 cancers-13-00433-t002:** Agents used in the clinic to stimulate immunogenic functions of DCs in cancer as monotherapy or in combination.

Agonist		Cancer Type(s)	Phase(s)	Interventions	Trials
FLT3L	CDX-301	Metastatic Breast Cancer, Head and Neck Squamous Cell Carcinoma	I/II	Radiation, Poly ICLC, Pembrolizumab	NCT03789097
	CDX-301	Non-Small Cell Lung Cancer	II	Radiation	NCT02839265
	CDX-301	Colorectal Cancer, Metastatic Cancer	I	-	NCT00003431
	Ad-hCMV-TK and Ad-hCMV-Flt3L	Malignant Glioma, Glioblastoma Multiforme	I	-	NCT01811992
	CDX-301	Non-Small Cell Lung Cancer Lung Cancer	I/II	Anti-CD40 Agonist Antibody, SBRT	NCT04491084
	CDX-301	Stage IV Melanoma, Stage IV Renal Cell Cancer, Recurrent Renal Cell Cancer, Recurrent Melanoma	II	gp100, MART-1, Montanide ISA-51 tyrosinase peptide	NCT00019396
	CDX-301	Kidney Cancer, Melanoma (Skin)	I	Recombinant CD40-ligand	NCT00020540
	CDX-301	Cutaneous, Mucosal and Ocular Melanoma	II	DEC-205/NY-ESO-1, Fusion Protein CDX-1401, Neoantigen-based, Melanoma-Poly-ICLC Vaccine	NCT02129075
	CDX-301	Melanoma, Non Small Cell Lung Cancer and others	I	CDX-1140, Pembrolizumab, Chemotherapy	NCT03329950
	CDX-301	Breast Cancer	I	Anti-CD40 Agonist, Poly ICLC, Radiation	NCT04616248
	CDX-301	Breast Cancer	I/II	Filgrastim, Thrombopoietin, Interleukin-3	NCT00006225
TLR2	CBLB612	Breast Cancer	II	-	NCT02778763
TLR4	GLA-SE	Colorectal Cancer Metastatic	I	FOLFOX, Nivolumab, Ipilimumab	NCT03982121
	GSK1795091	Neoplasms	I	GSK3174998, GSK3359609, Pembrolizumab	NCT03447314
	GSK1795091	Neoplasms	I	-	NCT02798978
	GLA-SE	Melanoma	I	MART-1 Antigen	NCT02320305
	GLA-SE	Soft Tissue Sarcoma	I	Radiation	NCT02180698
	GLA-SE	Merkel Cell Carcinoma	I	-	NCT02035657
	OM-174	Neoplasms	I	-	NCT01800812
TLR3	Hiltonol	Melanoma	I/II	NY-ESO-1 protein, Montanide	NCT01079741
	Hiltonol	Head and Neck Squamous Cell Carcinoma, Breast and others	I/II	Durvalumab, Tremelimumab	NCT02643303
	Hiltonol	Ovarian cancer and others	I	OC-L, Montanide	NCT02452775
	Hiltonol	Ovarian cancer	I	Oregovomab	NCT03162562
	Ampligen	Ovarian cancer	I/II	OC-L, Montanide, Prevnar	NCT01312389
	Hiltonol	Glioma	II	Autologous tumor lysate-pulsed DC vaccination, Tumor lysate-pulsed DC vaccination+0.2% resiquimod	NCT01204684
	Hiltonol	Metastatic colon cancer Neoplasms	I/II	Pembrolizumab	NCT02834052
	Hiltonol	Glioma	II	Bevacizumab, Peptide Vaccine, Poly-ICLC as immune adjuvant, Keyhole limpet hemocyanin	NCT02754362
TLR7	RO7119929	Hepatocellular Carcinoma, Biliary Tract Cancer, Secondary Liver Cancer, Liver Metastases	I	Tocilizumab	NCT04338685
	SHR2150	Neoplasms	I/II	Chemotherapy, PD1 Ab, CD47 Ab	NCT04588324
	Imiquimod (R837)	Breast Cancer	II	-	NCT00899574
		Melanoma and others	I	PD-1 Antibody Blockade	NCT04116320
		Breast Cancer	I/II	Cyclophosphamide, Radiation	NCT01421017
		High Grade Cervical Intraepithelial Neoplasia	I	Topical Fluorouracil	NCT03196180
	DSP-0509	Neoplasms	I/II	Pembrolizumab	NCT03416335
	MEDI9197	Neoplasms	I	Durvalumab	NCT02556463
	Resiquimod	Neoplasms	I	NY-ESO-1, Montanide ISA^®^-51 VG	NCT00821652
	852A	Breast Cancer and others	II	-	NCT00319748
	NJH395	NON-breast HER2+ Cancers	I	-	NCT03696771
	BNT411	Neoplasms	I/II	Atezolizumab, Carboplatin, Etoposide	NCT04101357
	TQ-A3334	Non-Small Cell Lung Cancer	I/II	Anlotinib	NCT04273815
	NKTR-262	Melanoma and others	I/II	Bempegaldesleukin, Nivolumab	NCT03435640
	BCDC-1001	Breast Cancer, Gastric Cancer	I/II	Pembrolizumab	NCT04278144
	LHC165 PDR001	Neoplasms	I	-	NCT03301896
TLR9	CpG	Pancreatic Cancer, Metastatic Pancreatic Cancer	I	Irreversible Electroporation, Nivolumab	NCT04612530
	CMP-001	Melanoma	II	Nivolumab, [18F]F-AraG PET/CT	NCT04401995
	CMP-001	Locally Advanced Malignant Solid Neoplasm, Metastatic Pancreatic Adenocarcinoma	I/II	Agonistic Anti-OX40	NCT04387071
	Tilsotolimod	Advanced Neoplasms	I	Ipilimumab, Nivolumab	NCT04270864
		Malignant Melanoma	II	-	NCT04126876
	SD-101	Metastatic Pancreatic Adenocarcinoma, Refractory Pancreatic Adenocarcinoma, Stage IV Pancreatic Cancer AJCC	I	Nivolumab, Radiation	
	SD-101	Advanced Malignant Solid Neoplasm, Extracranial Solid Neoplasm, Metastatic Malignant Solid Neoplasm	I	Anti-OX40 Antibody BMS 986178	NCT03831295
	CMP-001	Melanoma	II	Nivolumab	NCT03618641
	CMP-001	Colorectal Neoplasms Malignant, Liver Metastases	I	Radiation, Nivolumab, Ipilimumab	NCT03507699
	IMO-2125	Metastatic Melanoma	III	Ipilimumab	NCT03445533
	CMP-001	Advanced Cancers	II	Avelumab, Utomilumab, PF-04518600, PD 0360324	NCT02554812
	EMD 1201081	Head and Neck Squamous Cell Carcinoma	I	5-FU, Cisplatin, Cetuximab	NCT01360827
	EMD 1201081	Head and Neck Squamous Cell Carcinoma	II	Cetuximab	NCT01040832
	IMO-2055	Colorectal Cancer	I	Cetuximab, FOLFIRI	NCT00719199
	CpG-7909	Esophageal Cancer	I/II	URLC10-177, TTK-567	NCT00669292

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
