# Peer review of "Dendritic Cells: Behind the Scenes of T-Cell Infiltration into the Tumor Microenvironment"

_cancers, 2021, doi:10.3390/cancers13030433_

Round 1
Reviewer 1 Report
- Overall this is an interesting review.
- Recommend to change title to reflect the content of the paper
- Markers have very different roles in mice and human. CD103 is one such example. The authors should clearly state whether they are talking about mouse biology or human biology in each instance. It might be good to separate the two
- It would be useful to have a table to demonstrate how to identify the different subsets identified in the paper (mouse and human).
Author Response
The point by point response has been uploaded as a doc file

Reviewer 2 Report
In their manuscript Lucarini & Melaiu et al. describe the role of dendritic cells within solid tumors. Their review is systematic and comprehensive, and figures/tables outstanding (with the exception of the somewhat cumbersome Figure 2). Specific comments on how to improve this work are described below:
-Though their article is entitled “Behind the scenes of T-cell infiltration into the tumor microenvironment”, this is a review about dendritic cells in cancer, which is in any way reflected in the current title. Therefore, I strongly suggest that the authors retitle their work to more accurately reflect its content and improve visibility on PubMed, etc.
-While the authors do an admirable job describing DC based therapies that are under clinical investigation, they fail to mention a variety of approaches including the Autologous Dendritic Cell Vaccine (MESOVAX) under investigation in advanced, PD-L1+ Mesothelioma (NCT03546426).
-Several additional means of enhancing the cross presentation of tumor antigen have been explored both in basic science research and in clinical trial that have not been included in this review. These include radiation (mentioned briefly), chemotherapy (particularly platinum based), irreversible electroporation, and cyroablation. These warrant an expanded discussion, as each has significant effects on DC recruitment and sterilizing immunity.
-Immune checkpoints such as PD-L1 have been shown to directly modulate DC function. This warrants a discussion in light of the advent of ICIs in cancer therapy.
Minor Points:
-It is generally good practice to order numbered titles sequentially e.g. CCL2, 3, 4, 5 etc.
-The manuscript should be carefully proofread prior to resubmission, and page-long paragraphs broken up into smaller, more discrete parts.
Author Response

(The authors gave the same response as above.)

Round 2
Reviewer 1 Report
This is an interesting paper and the authors address the critiques.